# ENHANCING LLM'S INTERPRETABILITY FOR TIME SE-RIES VIA MULTI-LEVEL ALIGNED EMBEDDINGS

## ABSTRACT

The adaptation of large language models (LLMs) to time series forecasting poses unique challenges, as time series data is continuous in nature, while LLMs operate on discrete tokens. Despite the success of LLMs in natural language processing (NLP) and other structured domains, aligning time series data with language-based representations while maintaining both predictive accuracy and interpretability remains a significant hurdle. Existing methods have attempted to reprogram time series data into text-based forms, but these often fall short in delivering meaningful, interpretable results. In this paper, we propose a multi-level text alignment framework for time series forecasting using LLMs that not only improves prediction accuracy but also enhances the interpretability of time series representations. Our method decomposes time series into trend, seasonal, and residual components, which are then reprogrammed into component-specific text representations. We introduce a multi-level alignment mechanism, where component-specific embeddings are aligned with pre-trained word tokens, enabling more interpretable forecasts. Experiments on multiple datasets demonstrate that our method outperforms state-of-the-art models in accuracy while providing good interpretability.

## 1 INTRODUCTION

Time series forecasting, which involves predicting future values based on historical data, has numerous practical applications, such as demand planning, inventory optimization, energy load forecasting, and climate modeling (Gao et al., 2020; Li et al., 2022; Liu et al., 2023a; Dimri et al., 2020). Traditionally, these tasks demand substantial domain expertise and carefully designed models tailored to specific datasets. However, recent advancements in pre-trained large language models (LLMs), such as GPT-4 (Achiam et al., 2023) and LLaMA (Touvron et al., 2023), have achieved remarkable success in natural language processing (NLP) and demonstrated potential in handling complex, structured domains. This raises a compelling question: how can these powerful pre-trained LLMs be effectively adapted for time series forecasting?

LLMs, trained on vast and diverse text corpora, provide a powerful foundation for various downstream tasks, requiring only minimal task-specific prompt engineering or fine-tuning. This flexibility has sparked a growing interest in leveraging LLMs for time series analysis. For example, methods like Promptcast (Xue & Salim, 2023) and LLMTime (Gruver et al., 2024) reformulate numerical inputs and outputs into prompts, treating time series forecasting as a sentence-to-sentence task, which enables the direct application of LLMs. Meanwhile, approaches like TEMPO (Cao et al., 2024) and GPT4TS (Zhou et al., 2023) take a different route by fine-tuning pre-trained LLMs, modifying components such as the Add&Norm layers and positional embeddings, further demonstrating LLMs' adaptability for time series forecasting.

Despite their potential, the benefits of LLMs in time series forecasting depend on the effective alignment between time series data and natural language modalities. For instance, TEST (Sun et al., 2023) developed an encoder that aligns time series data to word embedding space through instance-wise, feature-wise, and text-prototype-aligned contrast. TimeLLM (Jin et al., 2024a) introduced a reprogramming framework that aligns time series patches with text prototypes, while $S^2$IP-LLM (Pan et al., 2024) employed a semantically informed prompt to bridge time series embeddings and semantic space. These approaches, however, primarily achieve a "time series→pattern→text" trans-

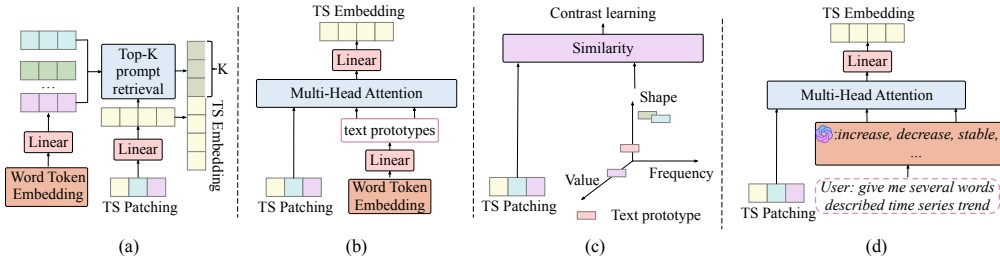

Figure 1: Cross-modality time series embeddings of $(a)$ semantic informed prompt, $(b)$ text prototypes reprogramming, $(c)$ contrast learning of text-prototype-aligned time series embeddings, and $(d)$ anchor alignment of our multi-level alignment.

formation to activate LLMs for time series tasks. This process often leads to unexpected outcomes. For example, the embedding of a subsequence with an upward trend may be misaligned with a word representing a decline or with a word that doesn't capture the trend at all. As a result, the challenge remains to fully unlock LLMs' capabilities for general time series forecasting in a way that is both accurate and interpretable.

In this paper, we address the challenge of interpretability in LLM-based time series forecasting by developing an interpretable multi-level text alignment framework while preserving the backbone model. Our approach consists of two key principles for effective time series representation learning: (a) modeling specific time series components such as trend, seasonality, and residuals, and (b) deriving interpretable explanations from the inherent properties of time series data through multi-level text alignment. Specifically, we decompose the time series input into three additive components—trend, seasonality, and residuals—using locally weighted scatterplot smoothing (LOESS) (Cleveland et al., 1990). These components are then reprogrammed into component-specific text representations that better align with the language capabilities of LLMs. Additionally, we employ component-specific prompts to guide the generation of learnable continuous vector representations that encode temporal knowledge of each component.

In summary, the main contributions of this paper are as follows: (1) We propose an interpretable multi-level text alignment framework for time series forecasting using LLMs, while keeping the backbone model unchanged. (2) Our method leverages this multi-level alignment to map decomposed time series components—trend, seasonality, and residuals—into distinctive, informative joint representations. The aligned trend-specific anchors enhance the interpretability of LLMs, while the aligned seasonality and residual prototypes improve the overall representation of the input time series. (3) Experimental results on multiple datasets validate the superiority of our model over state-of-the-art approaches, highlighting the effectiveness of interpretable multi-level text alignment.

## 2 RELATED WORK

### 2.1 PRE-TRAINED LARGE LANGUAGE MODELS FOR TIME SERIES.

The recent advancements in Large Language Models (LLMs) have opened up new opportunities for time series modeling. LLMs like T5 (Raffel et al., 2020), GPT-2 (Radford et al., 2019), GPT-4 (Achiam et al., 2023), and LLaMA (Touvron et al., 2023) have demonstrated impressive capabilities in understanding complex dependencies in heterogeneous textual data and generating meaningful outputs. Recently, there has been growing interest in exploring how to transfer the knowledge embedded in these pre-trained LLMs to the time series domain (Jin et al., 2024b; Jiang et al., 2024). For instance, (Xue & Salim, 2023) converts time series data into text sequences, achieving promising results. Other works, such as (Zhou et al., 2023; Gruver et al., 2024), tokenize time series data into overlapping patches and strategically fine-tune LLMs for time series forecasting tasks. Similarly, recent works such as (Cao et al., 2024; Pan et al., 2024) decompose time series data and use retrieval-based prompts to enhance fine-tuning of pre-trained LLMs. However, these approaches often fall short of delivering interpretable results and tend to treat time series as mere sequences of tokens, overlooking their inherent temporal structures. Converting numerical data to text without sufficient alignment to temporal dynamics can lead to inaccurate predictions and a lack of trans-

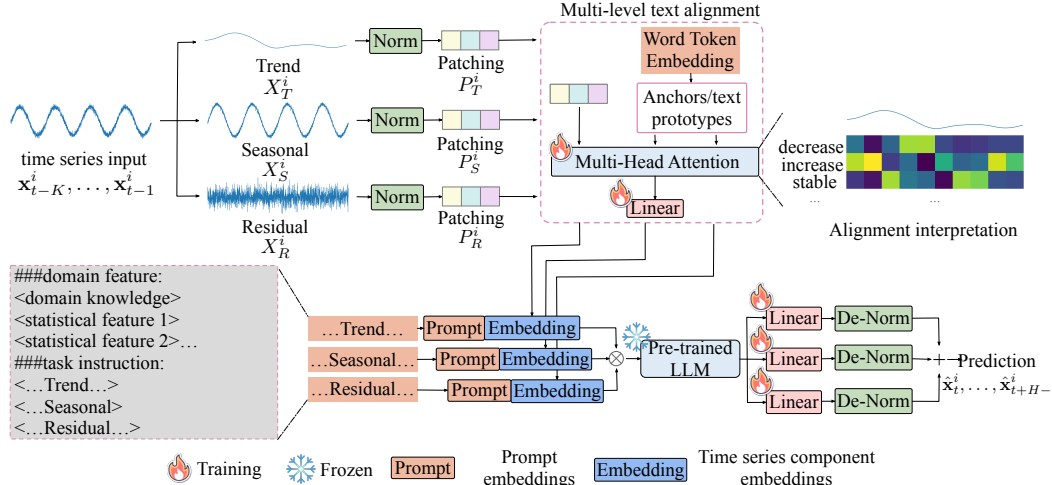

Figure 2: The architecture of the proposed multi-level aligned embeddings begins with the decomposition of the input time series into three components: trend, seasonal, and residual. These components tokenized and embedded are reprogrammed using anchors and condensed text prototypes to align the time series data with word tokens. Component-specific prefixed prompts are then added to guide the transformation of input patches. The outputs from the LLM are projected, de-normalized, and subsequently summed to generate the final prediction.

parency in the model's decision-making process, especially for multivariate time series. Our work introduces an interpretable multi-level text alignment framework to align time series components with language-based representations while keeping pre-trained LLMs intact.

## 2.2 TIME SERIES ALIGNED EMBEDDINGS

A key challenge in adapting LLMs for time series forecasting lies in aligning the continuous nature of time series data with the discrete token-based embeddings used in language models. Inspired by prototype-level contrast methods (Caron et al., 2020), (Sun et al., 2023) select certain text embeddings as basic prototypes to guide and constrain the learning of time series token embeddings. Similarly, (Jin et al., 2024a) reprogram time series data using the source data modality alongside prompts without modifying the input time series directly or fine-tuning the backbone LLM. These methods essentially follow a "time series→pattern→text" paradigm to activate LLMs for time series forecasting. However, the selection of text prototypes in these approaches is often arbitrary, and the chosen prototypes may not accurately reflect the underlying characteristics of the time series data (Sun et al., 2023). To address this limitation and enhance the interpretability of LLMs for time series, our approach selects time series-specific anchors to guide and constrain the learning of time series token embeddings. By aligning these embeddings with time series-related prototypes, we improve both the interpretability and performance of LLMs for time series forecasting.

## 3 METHODOLOGY

Our approach focuses on enhancing the interpretability of large language models (LLMs) for time series data through multi-level aligned embeddings. As illustrated in Figure 2, the proposed framework consists of four core modules: (1) time series input decomposition, (2) multi-level text alignment, (3) component-specific prompts, and (4) output projection. The process begins by partitioning a multivariate time series into $N$ univariate time series, each processed independently. The $i$-th series, denoted as $\mathbf{X}^{(i)} \in \mathbb{R}^{1 \times L}$, undergoes a series of steps including decomposition, normalization, patching, and embedding before being aligned with anchor points and text prototypes. To enhance the LLM's reasoning capability on time series data, we introduce component-specific prompts along with the aligned embeddings, enabling the model to generate meaningful output representations. These representations are then projected through an output linear layer to produce the final fore-

casts, $\hat{\mathbf{x}}_t^{(i)}, \ldots, \hat{\mathbf{x}}_{t+H-1}^{(i)}$. With the primary objective of improving interpretability, we utilize GPT-2 (Radford et al., 2019), employing its first six layers as the backbone model for time series forecasting without fine-tuning the foundational model.

## 3.1 PROBLEM STATEMENT

Given the observed values over the previous $L$ timestamps, the task of multivariate time series forecasting is to predict the values for the next $H$ timestamps. Formally, this can be represented as:

$$\hat{\mathbf{x}}_t^{(i)}, \ldots, \hat{\mathbf{x}}_{t+H-1}^{(i)} = F\left(\mathbf{x}_{t-L}^{(i)}, \ldots, \mathbf{x}_{t-1}^{(i)}; \mathbf{V}^{(i)}\right), \tag{1}$$

where $\hat{\mathbf{x}}_t^{(i)}, \ldots, \hat{\mathbf{x}}_{t+H-1}^{(i)}$ is the vector of $H$-step prediction from timestamp $t$ of channel $i$ corresponding to the $i$-th feature. Given the historical values $\mathbf{x}_{t-L}^{(i)}, \ldots, \mathbf{x}_{t-1}^{(i)}$, a large language model $F$ uses prompt $\mathbf{V}^{(i)}$ to make these predictions. Leveraging the strong reasoning capabilities of pretrained large language models, we aim to align time series data with text to enable LLMs to interpret the input series and accurately forecast the $H$ future steps, with the overall objective of minimizing the mean square errors between the ground truths and predictions, expressed as:

$$\frac{1}{H} \sum_{h=1}^{H} \|\hat{\mathbf{x}}_{t-1+h}^{(i)} - \mathbf{x}_{t-1+h}^{(i)}\|^2. \tag{2}$$

## 3.2 TIME SERIES INPUT DECOMPOSITION

For time series data, decomposing complex inputs into meaningful components such as trend, seasonal, and residual elements can help optimally extract valuable information. In this paper, given the input $X \in \mathbb{R}^{N \times L}$, where $N$ is the feature size and $L$ is the length of the time series, the additive decomposition can be represented as:

$$X^{(i)} = X_T^{(i)} + X_S^{(i)} + X_R^{(i)}, \tag{3}$$

where $i$ refers to the feature index for multivariate time series input. The trend component $X_T \in \mathbb{R}^{N \times L}$, capturing the underlying long-term patterns in the data, is expressed as $X_T = \frac{1}{m} \sum_{j=-k}^{k} X_{t+j}$, where $m = 2k + 1$ and $k$ is the averaging step size. The seasonal component $X_S \in \mathbb{R}^{N \times L}$ reflects the repeating short-term cycles and can be estimated after removing the trend. The residual component $X_R \in \mathbb{R}^{N \times L}$ represents the remainder of the data once the trend and seasonal elements have been extracted. There are multiple methods available for performing additive seasonal-trend decomposition. One common approach is the classical additive seasonal-trend decomposition, which first extracts the long-term trend using moving averages. The seasonal component is then estimated by averaging the detrended time series, and the residual is obtained by subtracting the estimated trend and seasonal components. Another widely used method is the Seasonal-Trend decomposition using Loess (STL) (Cleveland et al., 1990). The choice of decomposition method in this paper is determined based on validation results.

Following the approach outlined in (Nie et al., 2023), we patch the decomposed components of the time series. Specifically, for the $i$-th normalized trend component, we obtain the patched token $\mathbf{P}_T^{(i)} \in \mathbb{R}^{L_P \times K}$, where $L_P$ represents the patch length and $K = \lfloor \frac{(L-L_P)}{s} \rfloor + 2$ denotes the number of patches, with $s$ is the stride. Similarly, we apply this patching process to the seasonal and residual components, obtaining patched tokens $\mathbf{P}_S^{(i)}$ and $\mathbf{P}_R^{(i)}$, respectively. These patched tokens are then fed into the multi-level text alignment module to produce aligned time series embeddings.

## 3.3 MULTI-LEVEL TEXT ALIGNMENT

Here we reprogram patch embeddings into the LLMs' pre-training data representation space to align the modalities of time series and natural language to activate the backbone's time series understanding and reasoning capabilities. Naively, we can align the token embedding of time series and text using similarity estimation. Although time series tokens lack text annotation, we can place their embedding near typical text descriptions of time series. Thus, it is intuitively expected that various time series tokens can represent various descriptive words such as up, down, stable, and so on. However,

the pre-trained word token embedding space is vast and dense, and the selection of text prototypes (patterns) is often highly relaxed, sometimes even involving random words unrelated to time series or clusters of pre-trained word tokens (Sun et al., 2023; Pan et al., 2024; Jin et al., 2024a), which leads to poor interpretability.

In this work, we propose multi-level text alignment to enhance the interpretability of LLMs on time series forecasting. We first decompose the time series into trend, seasonal, and residual and align the trend $X_T$ with a selected trend-specific word pool $\mathbb{W}_{trend}$. Besides, we reprogram seasonal $X_S$ and residual $X_R$ using pre-trained word embeddings $\mathbf{E} \in \mathbb{R}^{V \times D}$ in the backbone, where $V$ is the vocabulary size, $D$ is the hidden dimension of the pre-trained LLM. Directly leveraging $\mathbf{E}$ will result in large and potentially dense reprogramming space. We adapt linearly probing $\mathbf{E}$. The text prototypes of seasonal and residual denoted as $\mathbf{E}'_{seasonal} \in \mathbb{R}^{V'_{seasonal} \times D}$ and $\mathbf{E}'_{residual} \in \mathbb{R}^{V'_{residual} \times D}$, where $V'_{seasonal} < V'_{residual} \ll V$ because the residual is more inconsistent and variable compared to the seasonal.

As illustrated in the top-right of Figure 2, our multi-level text alignment aims to give a connection between anchors and trend patches. The selected anchors are sparse. We reprogram seasonal and residual with text prototypes connecting time series patches with a more dense reprogramming space. To realize this, we employ a multi-head cross-attention layer for each component. Specifically, for $i-$ input feature, we define query matrices $\mathbf{Q}_T^{(i)} = \mathbf{P}_T^{(i)} \mathbf{W}_T^{Q(i)}$, key matrices $\mathbf{K}_T^{(i)} = \mathbb{W}_{trend} \mathbf{W}_T^{K(i)}$, value matrices $\mathbf{V}_T^{(i)} = \mathbb{W}_{trend} \mathbf{W}_T^{V(i)}$ for trend; query matrices $\mathbf{Q}_S^{(i)} = \mathbf{P}_S \mathbf{W}_S^{Q(i)}$, key matrices $\mathbf{K}_S^{(i)} = \mathbf{E}'_{seasonal} \mathbf{W}_S^{K(i)}$, value matrices $\mathbf{V}_S^{(i)} = \mathbf{E}'_{seasonal} \mathbf{W}_S^{V(i)}$ for seasonal; query matrices $\mathbf{Q}_R^{(i)} = \mathbf{P}_R \mathbf{W}_R^{Q(i)}$, key matrices $\mathbf{K}_R^{(i)} = \mathbf{E}'_{residual} \mathbf{W}_R^{K(i)}$, value matrices $\mathbf{V}_R^{(i)} = \mathbf{E}'_{residual} \mathbf{W}_R^{V(i)}$ for residual. Through multi-head attention, we reprogram each time series component. For example, the trend after multi-head attention is defined as:

$$\mathbf{Z}_T^{(i)} = \text{ATTENTION}(\mathbf{Q}_T^{(i)}, \mathbf{K}_T^{(i)}, \mathbf{V}_T^{(i)}) = \text{SOFTMAX}\left(\frac{\mathbf{Q}_T^{(i)} \mathbf{K}_T^{(i)\top}}{\sqrt{d_k}}\right) \mathbf{V}_T^{(i)}, \qquad (4)$$

where $d_k$ is the dimension of each head in the multi-head attention module. After the multi-head attention step, each component is linearly projected to align the hidden dimensions with the backbone model.

## 3.4 COMPONENT-SPECIFIC PROMPTS

Prompting techniques have proven highly effective across various tasks by leveraging task-specific knowledge encoded in prompts. This success stems from prompts providing a structured framework that aligns the model's output with desired objectives, improving accuracy and coherence. However, directly translating time series into natural language poses challenges, complicating the creation of instruction-following datasets and effective on-the-fly prompting (Xue & Salim, 2022). Recent advances show that prompts can enrich input context and guide the transformation of reprogrammed time series patches (Jin et al., 2024a). To leverage the semantic information in time series components, we propose a component-specific prefix prompting strategy. This includes three elements: dataset context, input statistical features, and component-specific task instructions for trend, seasonal, and residual components. For instance, the task description *'forecast the next 96 steps given the previous 512 steps [trend, seasonal, residual]'* serves as a template for our task instructions, which are then concatenated with the corresponding component data.

## 3.5 OUTPUT PROJECTION

After packing and forwarding the component-specific prompts and embeddings through the frozen backbone LLM, we retain the embedding for each component and apply a linear projection to the output representation. By denormalizing and summing these representations, we derive the final forecasts $\hat{\mathbf{x}}_t^{(i)}, \ldots, \hat{\mathbf{x}}_{t+H-1}^{(i)}$.

## 4 EXPERIMENTS

In our experiments, the proposed method outperforms state-of-the-art forecasting approaches across various benchmarks, including long-term, short-term, and few-shot forecasting. For a fair comparison, we follow the configurations outlined in (Wu et al., 2022) across all baselines, utilizing a unified evaluation pipeline[1]. Our code will be made available on GitHub upon the acceptance of the paper.

**Baselines.** We compare with the SOTA time series models and cite their performance from (Zhou et al., 2023; Chang et al., 2023) if applicable. The SOTA includes a set of Transformer-based methods, i.e., PatchTST(Nie et al., 2023), ETSformer(Woo et al., 2022), Non-Stationary Transformer(Liu et al., 2022), FEDformer(Zhou et al., 2022), Autoformer(Wu et al., 2021), Informer(Zhou et al., 2021), and Reformer(Kitaev et al., 2019). We also select a set of non-transformer-based techniques, i.e., DLinear(Zeng et al., 2023), TimesNet(Wu et al., 2022), N-BEATS (Oreshkin et al., 2020a), and LightTS(Zhang et al., 2022). Finally, four methods are based on LLMs, i.e., TimeLLM(Jin et al., 2024a), LLM4TS(Chang et al., 2023), GPT4TS(Zhou et al., 2023), and LLMTime(Gruver et al., 2024). Aligned with the GPT4TS configuration (Zhou et al., 2023), we utilize only the first 6 layers of the 12-layer GPT-2 base (Radford et al., 2019) as the backbone model of ours and TimeLLM.

### 4.1 LONG-TERM FORECASTING

**Setup.** For long-term forecasting, we evaluate on ETTh1, ETTh2, ETTm1, ETTm2, Weather, Electricity(ECL), and Traffic, which have been widely adopted as benchmarking datasets for long-term forecasting works (Wu et al., 2022). Details of these datasets are shown in Appendix A. The input time series length $L$ is set as 512, and we evaluate across four prediction horizons: $H \in \{96, 192, 336, 720\}$. The evaluation metrics include mean square error (MSE) and mean absolute error (MAE).

**Results.** Table 1 presents the performance of various time series forecasting models on MSE and MAE metrics across different prediction horizons on multiple benchmarks. Our proposed model consistently outperforms existing baselines, demonstrating superior performance on average across most datasets and prediction lengths. This highlights the broad applicability of multi-level text alignment. Notably, our comparison with TimeLLM—a recent work leveraging text prototype reprogramming to align time series with text tokens—is significant. Specifically, our model achieves substantial improvements on the Weather and ETTm1 datasets, exceeding the best-performing LLM-based model, LLM4TS, by **23.3%** and **26.8%**, respectively, in terms of MSE. Additionally, it records the lowest error rates across numerous individual dataset-prediction length configurations. These results suggest that integrating LLMs with multi-level text alignment can significantly enhance the accuracy of long-term time series forecasting.

### 4.2 FEW-SHOT FORECASTING

**Setups.** LLMs have recently shown impressive few-shot learning capabilities (Liu et al., 2023b). To evaluate performance in the few-shot forecasting setting, we follow the experimental setup outlined in (Zhou et al., 2023), allowing us to assess whether the model can generate accurate forecasts with limited training data. In these experiments, we use only the first 10% of the training data.

**Results.** The brief 10% few-shot learning results in Table 2 and full results in Appendix B.1 demonstrate that our model significantly outperforms all baseline methods across most datasets. We attribute this success to the effective knowledge activation achieved through multi-level text alignment. Specifically, our model improves few-shot learning performance on the Weather and Traffic datasets by **11.9%** and **5%**. When trained with only 10% of the data, LLM-based methods substantially outperform other baselines, which are trained from scratch and thus limited by the smaller training set. In contrast, LLM-based models can leverage pre-trained knowledge and align it with time series embeddings to enhance representation.

---

[1]https://github.com/thuml/Time-Series-Library

Table 1: Long-term forecasting results for {96, 192, 336, 720} horizons. Lower values indicate better performance. Red: the best, Blue: second best.

| Methods | | Ours | | Time-LLM | | LLM4TS | | GPT4TS | | DLinear | | PatchTST | |
|---|---|---|---|---|---|---|---|---|---|---|---|---|---|
| Datasets \ Horizon | | MSE | MAE | MSE | MAE | MSE | MAE | MSE | MAE | MSE | MAE | MSE | MAE |
| ETTh1 | 96 | 0.355 | 0.404 | 0.384 | 0.407 | 0.371 | 0.394 | 0.376 | 0.397 | 0.375 | 0.399 | 0.370 | 0.399 |
| | 192 | 0.426 | 0.445 | 0.423 | 0.434 | 0.403 | 0.412 | 0.416 | 0.418 | 0.405 | 0.416 | 0.413 | 0.421 |
| | 336 | 0.434 | 0.449 | 0.435 | 0.447 | 0.420 | 0.422 | 0.442 | 0.433 | 0.439 | 0.443 | 0.422 | 0.436 |
| | 720 | 0.480 | 0.493 | 0.439 | 0.463 | 0.422 | 0.444 | 0.477 | 0.456 | 0.472 | 0.490 | 0.447 | 0.466 |
| | Avg. | 0.424 | 0.448 | 0.420 | 0.438 | 0.404 | 0.418 | 0.428 | 0.426 | 0.422 | 0.437 | 0.413 | 0.435 |
| ETTh2 | 96 | 0.260 | 0.336 | 0.295 | 0.355 | 0.269 | 0.332 | 0.285 | 0.342 | 0.289 | 0.353 | 0.274 | 0.336 |
| | 192 | 0.333 | 0.375 | 0.376 | 0.410 | 0.328 | 0.377 | 0.354 | 0.389 | 0.383 | 0.418 | 0.339 | 0.379 |
| | 336 | 0.369 | 0.408 | 0.376 | 0.412 | 0.353 | 0.396 | 0.373 | 0.407 | 0.448 | 0.465 | 0.329 | 0.380 |
| | 720 | 0.444 | 0.455 | 0.410 | 0.442 | 0.383 | 0.425 | 0.406 | 0.441 | 0.605 | 0.551 | 0.379 | 0.422 |
| | Avg. | 0.378 | 0.408 | 0.364 | 0.403 | 0.333 | 0.383 | 0.355 | 0.394 | 0.431 | 0.446 | 0.330 | 0.379 |
| ETTm1 | 96 | 0.117 | 0.232 | 0.297 | 0.349 | 0.285 | 0.343 | 0.292 | 0.346 | 0.299 | 0.343 | 0.290 | 0.342 |
| | 192 | 0.198 | 0.298 | 0.336 | 0.373 | 0.324 | 0.366 | 0.332 | 0.372 | 0.335 | 0.365 | 0.332 | 0.369 |
| | 336 | 0.301 | 0.360 | 0.362 | 0.390 | 0.353 | 0.385 | 0.366 | 0.394 | 0.369 | 0.386 | 0.366 | 0.392 |
| | 720 | 0.389 | 0.411 | 0.410 | 0.421 | 0.408 | 0.419 | 0.417 | 0.421 | 0.425 | 0.421 | 0.416 | 0.425 |
| | Avg. | 0.251 | 0.325 | 0.351 | 0.383 | 0.343 | 0.378 | 0.352 | 0.383 | 0.357 | 0.378 | 0.351 | 0.380 |
| ETTm2 | 96 | 0.095 | 0.200 | 0.177 | 0.264 | 0.165 | 0.254 | 0.173 | 0.262 | 0.167 | 0.269 | 0.165 | 0.255 |
| | 192 | 0.174 | 0.263 | 0.253 | 0.312 | 0.220 | 0.292 | 0.229 | 0.301 | 0.224 | 0.303 | 0.220 | 0.292 |
| | 336 | 0.243 | 0.313 | 0.285 | 0.345 | 0.268 | 0.326 | 0.286 | 0.341 | 0.281 | 0.342 | 0.274 | 0.329 |
| | 720 | 0.343 | 0.380 | 0.366 | 0.390 | 0.350 | 0.380 | 0.378 | 0.401 | 0.297 | 0.421 | 0.362 | 0.385 |
| | Avg. | 0.214 | 0.289 | 0.270 | 0.328 | 0.251 | 0.313 | 0.267 | 0.326 | 0.267 | 0.333 | 0.255 | 0.315 |
| Weather | 96 | 0.059 | 0.125 | 0.158 | 0.210 | 0.147 | 0.196 | 0.162 | 0.212 | 0.176 | 0.237 | 0.149 | 0.198 |
| | 192 | 0.115 | 0.188 | 0.191 | 0.240 | 0.191 | 0.238 | 0.204 | 0.248 | 0.220 | 0.282 | 0.194 | 0.241 |
| | 336 | 0.211 | 0.263 | 0.247 | 0.284 | 0.241 | 0.277 | 0.254 | 0.286 | 0.265 | 0.319 | 0.245 | 0.282 |
| | 720 | 0.299 | 0.327 | 0.319 | 0.334 | 0.313 | 0.329 | 0.326 | 0.337 | 0.333 | 0.362 | 0.314 | 0.334 |
| | Avg. | 0.171 | 0.226 | 0.229 | 0.267 | 0.223 | 0.260 | 0.237 | 0.271 | 0.248 | 0.300 | 0.225 | 0.264 |
| ECL | 96 | 0.116 | 0.221 | 0.137 | 0.237 | 0.128 | 0.223 | 0.139 | 0.238 | 0.140 | 0.237 | 0.129 | 0.222 |
| | 192 | 0.145 | 0.250 | 0.150 | 0.249 | 0.146 | 0.240 | 0.153 | 0.251 | 0.153 | 0.249 | 0.150 | 0.240 |
| | 336 | 0.167 | 0.271 | 0.168 | 0.266 | 0.163 | 0.258 | 0.169 | 0.266 | 0.169 | 0.267 | 0.163 | 0.259 |
| | 720 | 0.209 | 0.307 | 0.203 | 0.293 | 0.200 | 0.292 | 0.206 | 0.297 | 0.203 | 0.301 | 0.197 | 0.290 |
| | Avg. | 0.159 | 0.262 | 0.164 | 0.261 | 0.159 | 0.253 | 0.167 | 0.263 | 0.166 | 0.263 | 0.161 | 0.252 |
| Traffic | 96 | 0.255 | 0.229 | 0.380 | 0.277 | 0.372 | 0.259 | 0.388 | 0.282 | 0.410 | 0.282 | 0.360 | 0.249 |
| | 192 | 0.332 | 0.258 | 0.399 | 0.288 | 0.391 | 0.265 | 0.407 | 0.290 | 0.423 | 0.287 | 0.379 | 0.256 |
| | 336 | 0.370 | 0.273 | 0.408 | 0.290 | 0.405 | 0.275 | 0.412 | 0.294 | 0.436 | 0.296 | 0.392 | 0.264 |
| | 720 | 0.428 | 0.301 | 0.445 | 0.308 | 0.437 | 0.292 | 0.450 | 0.312 | 0.466 | 0.315 | 0.432 | 0.286 |
| | Avg. | 0.346 | 0.265 | 0.408 | 0.290 | 0.401 | 0.273 | 0.414 | 0.294 | 0.433 | 0.295 | 0.390 | 0.263 |
| 1st Count | | 26 | | 0 | | 9 | | 0 | | 0 | | 9 | |

Table 2: Few-shot learning on 10% training data. All results are averaged from four different forecasting horizons: $H \in \{96, 192, 336, 720\}$. Lower values indicate better performance.

| Methods | Ours | | TimeLLM | | LLM4TS | | GPT4TS | | DLinear | | PatchTST | |
|---|---|---|---|---|---|---|---|---|---|---|---|---|
| Metric | MSE | MAE | MSE | MAE | MSE | MAE | MSE | MAE | MSE | MAE | MSE | MAE |
| ETTh2 | 0.397 | 0.431 | 0.446 | 0.464 | 0.366 | 0.407 | 0.397 | 0.421 | 0.605 | 0.538 | 0.415 | 0.431 |
| ETTm2 | 0.262 | 0.324 | 0.292 | 0.343 | 0.276 | 0.324 | 0.293 | 0.335 | 0.316 | 0.368 | 0.296 | 0.343 |
| Weather | 0.207 | 0.263 | 0.359 | 0.275 | 0.235 | 0.270 | 0.238 | 0.275 | 0.241 | 0.283 | 0.242 | 0.279 |
| ECL | 0.190 | 0.288 | 0.182 | 0.277 | 0.172 | 0.264 | 0.176 | 0.269 | 0.180 | 0.280 | 0.180 | 0.273 |
| Traffic | 0.409 | 0.310 | 0.438 | 0.312 | 0.432 | 0.303 | 0.440 | 0.310 | 0.447 | 0.313 | 0.430 | 0.305 |
| 1st Count | 3 | | 0 | | 2 | | 0 | | 0 | | 1 | |

## 4.3 ZERO-SHOT FORECASTING

**Setups.** Beyond few-shot learning, LLMs also show promise as effective zero-shot learners. In this section, we evaluate the zero-shot learning capabilities of the multi-level text-aligned LLM. Specifically, we assess how well the model performs on one dataset after being optimized on another. Similar to the few-shot learning setup, we use the long-term forecasting protocol and evaluate various cross-domain scenarios utilizing the ETT datasets.

Table 3: Zero-shot learning results on ETT datasets. Lower values indicate better performance. Red: the best, Blue: second best.

| Methods | Ours | | Time-LLM | | GPT4TS | | LLMTime | | PatchTST | | DLinear | |
|---|---|---|---|---|---|---|---|---|---|---|---|---|
| Datasets | MSE | MAE | MSE | MAE | MSE | MAE | MSE | MAE | MSE | MAE | MSE | MAE |
| ETTh1 → ETTh2 | 0.346 | 0.396 | 0.354 | 0.400 | 0.406 | 0.422 | 0.992 | 0.708 | 0.380 | 0.405 | 0.493 | 0.488 |
| ETTh1 → ETTm2 | 0.294 | 0.357 | 0.310 | 0.363 | 0.325 | 0.363 | 1.867 | 0.869 | 0.314 | 0.360 | 0.415 | 0.452 |
| ETTh2 → ETTm2 | 0.276 | 0.345 | 0.303 | 0.356 | 0.335 | 0.370 | 1.867 | 0.869 | 0.325 | 0.365 | 0.328 | 0.386 |
| ETTm1 → ETTm2 | 0.217 | 0.284 | 0.275 | 0.325 | 0.313 | 0.348 | 1.867 | 0.869 | 0.296 | 0.334 | 0.335 | 0.389 |
| ETTm2 → ETTm1 | 0.562 | 0.478 | 0.501 | 0.453 | 0.769 | 0.567 | 1.933 | 0.984 | 0.568 | 0.492 | 0.649 | 0.537 |

Table 4: Full short-term time series forecasting. The forecasting horizons are in [6,48] and details are in Table 6. Lower values indicate better performance. Red: the best, Blue: second best.

| Methods | | Ours | Time-LLM | GPT4TS | TimesNet | PatchTST | N-BEATS | ETSformer | LightTS | DLinear | FEDformer |
|---|---|---|---|---|---|---|---|---|---|---|---|
| Average | SMAPE | 12.249 | 13.113 | 12.69 | 12.88 | 12.059 | 12.25 | 14.718 | 13.525 | 13.639 | 13.16 |
| | MASE | 1.678 | 1.77 | 1.808 | 1.836 | 1.623 | 1.698 | 2.408 | 2.111 | 2.095 | 1.775 |
| | OWA | 0.89 | 0.946 | 0.94 | 0.955 | 0.869 | 0.896 | 1.172 | 1.051 | 1.051 | 0.949 |

**Results.** The brief results are presented in Table 3, with full results in Appendix B.3. Our model demonstrates performance that is comparable to or surpasses other baselines. In data-scarce scenarios, our model significantly outperforms other LLM-based models, consistently providing better forecasts. Both our model and TimeLLM (Jin et al., 2024a) outperform traditional baselines, likely due to cross-modality alignment, which more effectively activates LLMs' knowledge transfer and reasoning capabilities for time series tasks. Additionally, our multi-level aligned embeddings better align language cues with temporal components of time series, enabling superior zero-shot forecasting performance compared to TimeLLM.

## 4.4 SHORT-TERM FORECASTING

**Setups.** We select the M4 benchmark as our testbed, which consists of a collection of marketing data with varying sampling frequencies. More details are provided in Appendix A.1. The prediction horizons in this case are relatively short, ranging from 6 to 48, with input lengths set to twice the prediction horizon. The evaluation metrics include symmetric mean absolute percentage error (SMAPE), mean absolute scaled error (MSAE), and overall weighted average (OWA).

**Results.** The brief results are presented in Table 4, with full results in Appendix B.2. Our method consistently surpasses TimeLLM and GPT4TS by **6.5%** and **3.5%**, respectively, and remains competitive with the state-of-the-art (SOTA) method, PatchTST (Nie et al., 2023).

## 4.5 MODEL ANALYSIS

Table 5: Performance comparison of different variants for long-term and few-shot forecasting.

| Variant | Long-term Forecasting | | Few-shot Forecasting | |
|---|---|---|---|---|
| | ETTm1-96 | ETThm1-192 | ETTm1-96 | ETTm1-192 |
| **Default** GPT-2 (6) | 0.117 | 0.198 | 0.360 | 0.429 |
| **A.1** w/o alignment | 0.262 | 0.347 | 0.571 | 0.583 |
| **B.1** only trend alignment | 0.184 | 0.283 | 0.476 | 0.578 |
| **B.2** only seasonal alignment | 0.127 | 0.212 | 0.367 | 0.432 |
| **B.3** only residual alignment | 0.171 | 0.229 | 0.433 | 0.506 |
| **C.1** noise anchors | 0.134 | 0.214 | 0.424 | 0.464 |
| **C.2** synonymous anchors | 0.119 | 0.202 | 0.366 | 0.434 |
| **D.1** w/o component-specific instruction | 0.125 | 0.205 | 0.408 | 0.461 |
| **D.2** w/o domain features | 0.118 | 0.199 | 0.371 | 0.440 |

**Multi-level text alignment variants.** Our results in Table 5 show that removing component alignment or prefixed prompts negatively impacts knowledge transfer during LLM reprogramming for effective time series forecasting. Specifically, without alignment (**A.1**), we observe a significant average performance drop of **75.4%** across standard and few-shot forecasting tasks. We also examine the effect of aligning only two components to assess whether aligning just one component is sufficient in our multi-level alignment strategy. Retaining only seasonal alignment (**B.2**) achieves the best performance, though it still results in an average MSE increase of **4.5%** across all scenarios. In contrast, keeping only trend alignment significantly degrades performance, with over **32.2%** performance loss in both standard and few-shot tasks. Furthermore, altering the selection of anchors for trend alignment (**C.1, C.2**) increases MSE by over **14.5%** when using noise anchors. Expanding the anchor selection with synonymous words produces results comparable to the default, with less than a **2%** variation in MSE. Finally, removing component-specific instruction (**D.1**) and domain features (**D.2**) results in MSE increases of **7.7%** and **1.7%**, respectively.

**Multi-level text alignment interpretation.** We present a case study on ETTm1 using non-overlapping patching, where the patch stride is the same as the patch length, with different selected anchors shown in Figure 3. The attention map illustrates the optimized attention scores between input trend patches and aligned anchors. These matched anchors serve as textual shapelets for the time series tokens. Specifically, subplot **(a)** displays the optimized attention scores for synonymous anchors, which are consistent. The highlighted anchors—"rise," "increase," "climb," "grow," and "expand"—are associated with upward trend patches. In contrast, subplot **(b)** shows no highlighted anchor when all trend patches are aligned with noise words unrelated to time series trends. This case study demonstrates that aligned anchors effectively summarize the textual shapelets of the input trend patches.

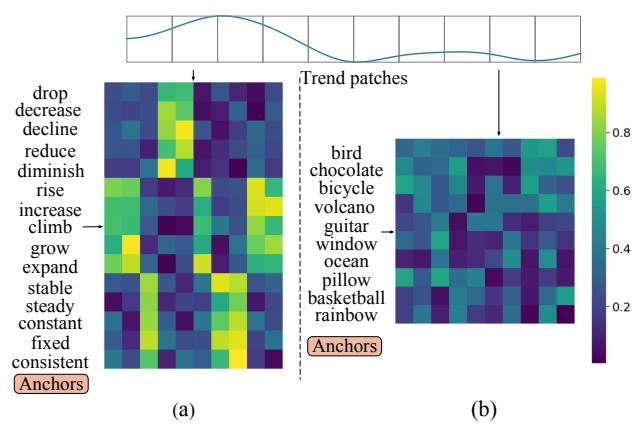

Figure 3: A showcase of visualization of multi-level alignment interpretation.

Although multi-level alignment in both seasonal and residual components can provide visual interpretations, visualizing these alignments is challenging. Since the input patches are aligned with text prototypes learned from a large and dense pre-trained word embedding space, more tools are needed to present a better visualization across two optimized layers. Moreover, the trend component is the most interpretable and semantically clear of the three components, in contrast to the noisy residual and the seasonal component, which lacks textual semantics. Our model efficiencies in terms of parameters, memory, and speed are comparable to TimeLLM (Jin et al., 2024a) with only two additional lightweight aligned embedding layers for integrating trend and seasonal components.

## 5 CONCLUSION AND FUTURE WORK

We propose a multi-level text alignment framework for time series forecasting utilizing pre-trained language models. Our multi-level aligned embeddings enhance the LLM's interpretability and forecasting performance by aligning time series components with anchors and text prototypes. Due to the impractical interpretability of the text prototypes aligned with the seasonal and residual components of the time series, and the sparsity of text prototypes describing the time series trend, we align the time series trend with selected anchors from the pre-trained word embeddings. Our results demonstrate that time series tokens aligned with anchors provide a clearer and more intuitive interpretation of similar time series trends. Future research should focus on optimizing the alignment module for selected anchors and time series tokens, and work toward developing multimodal models capable of joint reasoning across time series, natural language, and other modalities.

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

# A EXPERIMENTAL DETAILS

## A.1 IMPLEMENTATION DETAILS

Dataset statistics are summarized in Table 6. We evaluate the long-term forecasting performance on the well-established eight different benchmarks over different domains, including four ETT datasets (Zhou et al., 2021) (i.e., ETTh1, ETTh2, ETTm1, and ETTm2), Weather, Electricity, Traffic, and ILI from (Zhou et al., 2023). Furthermore, we evaluate the performance of short-term forecasting on the M4 benchmark (Makridakis et al., 2018).

Table 6: Datasets statistics are from (Zhou et al., 2023). The dataset size is organized in (training, validation, testing), and the dimension indicates the number of channels of time series.

| Tasks | Dataset | Dim. | Series Length | Dataset Size | Frequency | Domain |
|---|---|---|---|---|---|---|
| Long-term Forecasting | ETTm1 | 7 | {96, 192, 336, 720} | (34465, 11521, 11521) | 15 min | Temperature |
| | ETTm2 | 7 | {96, 192, 336, 720} | (34465, 11521, 11521) | 15 min | Temperature |
| | ETTh1 | 7 | {96, 192, 336, 720} | (8545, 2881, 2881) | 1 hour | Temperature |
| | ETTh2 | 7 | {96, 192, 336, 720} | (8545, 2881, 2881) | 1 hour | Temperature |
| | Electricity | 321 | {96, 192, 336, 720} | (18317, 2633, 5261) | 1 hour | Electricity |
| | Traffic | 862 | {96, 192, 336, 720} | (12185, 1757, 3509) | 1 hour | Transportation |
| | Weather | 21 | {96, 192, 336, 720} | (36792, 5271, 10540) | 10 min | Weather |
| Short-term Forecasting | M4-Yearly | 1 | 6 | (23000, 0, 23000) | Yearly | Demographic |
| | M4-Quarterly | 1 | 8 | (24000, 0, 24000) | Quarterly | Finance |
| | M4-Monthly | 1 | 18 | (48000, 0, 48000) | Monthly | Industry |
| | M4-Weekly | 1 | 13 | (359, 0, 359) | Weekly | Macro |
| | M4-Daily | 1 | 14 | (4227, 0, 4227) | Daily | Micro |
| | M4-Hourly | 1 | 48 | (414, 0, 414) | Hourly | Other |

The Electricity Transformer Temperature (ETT; An indicator reflective of long-term electric power deployment) benchmark is comprised of two years of data, sourced from two counties in China, and is subdivided into four distinct datasets, each with varying sampling rates: ETTh1 and ETTh2, which are sampled at a 1-hour level, and ETTm1 and ETTm2, which are sampled at a 15-minute level. Each entry within the ETT datasets includes six power load features and a target variable, termed "oil temperature". The Electricity dataset comprises records of electricity consumption from 321 customers, measured at a 1-hour sampling rate. The Weather dataset includes one-year records from 21 meteorological stations located in Germany, with a sampling rate of 10 minutes. The Traffic dataset includes data on the occupancy rates of the freeway system, recorded from 862 sensors across the State of California, with a sampling rate of 1 hour.

The M4 benchmark comprises 100K time series, amassed from various domains commonly present in business, financial, and economic forecasting. These time series have been partitioned into six distinctive datasets, each with varying sampling frequencies that range from yearly to hourly. These series are categorized into five different domains: demographic, micro, macro, industry, and finance.

## A.2 EVALUATION METRICS

For evaluation metrics, we utilize the mean square error (MSE) and mean absolute error (MAE) for long-term forecasting. In terms of the short-term forecasting on M4 benchmark, we adopt the symmetric mean absolute percentage error (SMAPE), mean absolute scaled error (MASE), and overall

weighted average (OWA) as in N-BEATS(Oreshkin et al., 2020b). Note that OWA is a specific metric utilized in the M4 competition. The calculations of these metrics are as follows:

$$\text{MSE} = \frac{1}{H} \sum_{h=1}^{H} \left( \mathbf{Y}_h - \hat{\mathbf{Y}}_h \right)^2, \qquad \text{MAE} = \frac{1}{H} \sum_{h=1}^{H} \left| \mathbf{Y}_h - \hat{\mathbf{Y}}_h \right|,$$

$$\text{SMAPE} = \frac{200}{H} \sum_{h=1}^{H} \frac{\left| \mathbf{Y}_h - \hat{\mathbf{Y}}_h \right|}{|\mathbf{Y}_h| + \left| \hat{\mathbf{Y}}_h \right|}, \qquad \text{MAPE} = \frac{100}{H} \sum_{h=1}^{H} \frac{\left| \mathbf{Y}_h - \hat{\mathbf{Y}}_h \right|}{|\mathbf{Y}_h|},$$

$$\text{MASE} = \frac{1}{H} \sum_{h=1}^{H} \frac{\left| \mathbf{Y}_h - \hat{\mathbf{Y}}_h \right|}{\frac{1}{H-s} \sum_{j=s+1}^{H} |\mathbf{Y}_j - \mathbf{Y}_{j-s}|}, \qquad \text{OWA} = \frac{1}{2} \left[ \frac{\text{SMAPE}}{\text{SMAPE}_{\text{Naïve2}}} + \frac{\text{MASE}}{\text{MASE}_{\text{Naïve2}}} \right],$$

where $s$ is the periodicity of the time series data. $H$ denotes the number of data points(i.e., prediction horizon in our cases). $\mathbf{Y}_h$ and $\hat{\mathbf{Y}}_h$ are the $h-$th ground truth and prediction where $h \in \{1, ..., H\}$.

### A.3 MODEL CONFIGURATIONS

The configurations of our models, relative to varied tasks and datasets, are consolidated in Table 7. By default, the Adam optimizer (Diederik, 2015) is employed throughout all experiments.

Table 7: An overview of the experimental configurations.

| Task-Dataset / Configuration | Model Hyperparameter | | | | | Training Process | | | |
|---|---|---|---|---|---|---|---|---|---|
| | $\mathbb{W}_{trend}$ | $V_{season}$ | $V_{resid}$ | Input Length $T$ | Patch Dim. $d_m$ | LR* | Loss | Batch Size | Epochs |
| LTF - ETTh1 | 10 | 100 | 1000 | 512 | 16 | $10^{-4}$ | MSE | 48 | 50 |
| LTF - ETTh2 | 10 | 100 | 1000 | 512 | 16 | $10^{-4}$ | MSE | 48 | 50 |
| LTF - ETTm1 | 10 | 100 | 1000 | 512 | 16 | $10^{-4}$ | MSE | 48 | 50 |
| LTF - ETTm2 | 10 | 100 | 1000 | 512 | 16 | $10^{-4}$ | MSE | 48 | 50 |
| LTF - Weather | 10 | 100 | 1000 | 512 | 16 | $10^{-4}$ | MSE | 24 | 50 |
| LTF - Electricity | 10 | 100 | 1000 | 512 | 16 | $10^{-4}$ | MSE | 24 | 50 |
| LTF - Traffic | 10 | 100 | 1000 | 512 | 16 | $10^{-4}$ | MSE | 24 | 50 |
| LTF - ILI | 10 | 100 | 1000 | 512 | 16 | $10^{-4}$ | MSE | 48 | 50 |
| STF - M4 | 10 | 100 | 1000 | $2 \times$H† | 16 | $10^{-4}$ | SMAPE | 48 | 50 |

†$H$ represents the forecasting horizon of the M4 datasets.
*LR represents the initial learning rate.

## B    FULL RESULTS OF ALL CONFIGURATIONS

### B.1    FEW-SHOT FORECASTING

Our full results in few-shot forecasting tasks are detailed in Table 8. With the scope of 10% few-shot learning, our model's secure SOTA performance in 13 out of 35 cases, spanning seven different time series benchmarks. Moreover, our model only lose to LLM4TS, which is neither interpretable nor light-weight.

### B.2    SHORT-TERM FORECASTING

Our complete results on short-term forecasting are presented in Table B.2. Our model consistently outperforms the majority of baseline models in most cases. Notably, we surpass TimeLLM by a large margin (e.g, **6.5%** average, **9.4%** on M4-Monthly), as well as GPT4TS (e.g., **10.5%** on M4-Yearly, **3.5%** on average). Compared to the state-of-the-art models, TimesNet,and PatchTST, our model exhibits comparable or superior performances without any parameter updates on the backbone LLM.

### B.3    ZERO-SHOT FORECASTING

The full results of zero-shot forecasting are summarized in Table 10. Our model remarkably surpasses the five most competitive time series models in zero-shot adaptation. Overall, we observe

Table 8: Full few-shot results on 10% training data.

| Methods | | Ours | | Time-LLM | | LLM4TS | | GPT4TS | | DLinear | | PatchTST | |
|---|---|---|---|---|---|---|---|---|---|---|---|---|---|
| Datasets \ Horizon | | MSE | MAE | MSE | MAE | MSE | MAE | MSE | MAE | MSE | MAE | MSE | MAE |
| ETTh1 | 96 | 0.629 | 0.548 | 0.530 | 0.492 | 0.417 | 0.432 | 0.458 | 0.456 | 0.492 | 0.495 | 0.516 | 0.485 |
| | 192 | 0.720 | 0.534 | 0.671 | 0.546 | 0.469 | 0.468 | 0.570 | 0.516 | 0.565 | 0.538 | 0.598 | 0.524 |
| | 336 | 0.893 | 0.622 | 0.907 | 0.639 | 0.505 | 0.499 | 0.608 | 0.535 | 0.721 | 0.622 | 0.657 | 0.550 |
| | 720 | 1.210 | 0.772 | 0.917 | 0.647 | 0.708 | 0.572 | 0.725 | 0.591 | 0.986 | 0.743 | 0.762 | 0.610 |
| | Avg. | 0.865 | 0.619 | 0.756 | 0.581 | 0.525 | 0.493 | 0.590 | 0.525 | 0.691 | 0.600 | 0.633 | 0.542 |
| ETTh2 | 96 | 0.330 | 0.383 | 0.349 | 0.418 | 0.282 | 0.351 | 0.331 | 0.374 | 0.357 | 0.411 | 0.353 | 0.389 |
| | 192 | 0.357 | 0.408 | 0.406 | 0.428 | 0.364 | 0.400 | 0.402 | 0.411 | 0.569 | 0.519 | 0.403 | 0.414 |
| | 336 | 0.406 | 0.439 | 0.488 | 0.489 | 0.374 | 0.416 | 0.406 | 0.433 | 0.671 | 0.572 | 0.426 | 0.441 |
| | 720 | 0.498 | 0.496 | 0.540 | 0.520 | 0.445 | 0.461 | 0.449 | 0.464 | 0.824 | 0.648 | 0.477 | 0.480 |
| | Avg. | 0.397 | 0.431 | 0.446 | 0.464 | 0.366 | 0.407 | 0.397 | 0.421 | 0.605 | 0.538 | 0.415 | 0.431 |
| ETTm1 | 96 | 0.360 | 0.389 | 0.297 | 0.349 | 0.360 | 0.388 | 0.390 | 0.404 | 0.352 | 0.392 | 0.410 | 0.419 |
| | 192 | 0.429 | 0.431 | 0.336 | 0.373 | 0.386 | 0.401 | 0.429 | 0.423 | 0.382 | 0.412 | 0.437 | 0.434 |
| | 336 | 0.446 | 0.465 | 0.362 | 0.390 | 0.415 | 0.417 | 0.469 | 0.439 | 0.419 | 0.434 | 0.476 | 0.454 |
| | 720 | 0.489 | 0.495 | 0.410 | 0.421 | 0.470 | 0.445 | 0.569 | 0.498 | 0.490 | 0.477 | 0.681 | 0.556 |
| | Avg. | 0.431 | 0.445 | 0.351 | 0.383 | 0.402 | 0.457 | 0.464 | 0.441 | 0.411 | 0.429 | 0.501 | 0.466 |
| ETTm2 | 96 | 0.126 | 0.231 | 0.192 | 0.276 | 0.184 | 0.265 | 0.188 | 0.269 | 0.213 | 0.303 | 0.191 | 0.274 |
| | 192 | 0.223 | 0.300 | 0.266 | 0.320 | 0.240 | 0.301 | 0.251 | 0.309 | 0.278 | 0.345 | 0.252 | 0.317 |
| | 336 | 0.290 | 0.345 | 0.317 | 0.356 | 0.294 | 0.337 | 0.307 | 0.346 | 0.338 | 0.385 | 0.306 | 0.353 |
| | 720 | 0.412 | 0.420 | 0.418 | 0.420 | 0.386 | 0.393 | 0.426 | 0.417 | 0.436 | 0.440 | 0.433 | 0.427 |
| | Avg. | 0.262 | 0.324 | 0.292 | 0.343 | 0.276 | 0.324 | 0.293 | 0.335 | 0.316 | 0.368 | 0.296 | 0.343 |
| Weather | 96 | 0.102 | 0.177 | 0.164 | 0.220 | 0.158 | 0.207 | 0.163 | 0.215 | 0.171 | 0.224 | 0.165 | 0.215 |
| | 192 | 0.164 | 0.234 | 0.215 | 0.258 | 0.204 | 0.249 | 0.210 | 0.254 | 0.215 | 0.263 | 0.210 | 0.257 |
| | 336 | 0.230 | 0.281 | 0.259 | 0.294 | 0.254 | 0.288 | 0.256 | 0.292 | 0.258 | 0.299 | 0.259 | 0.297 |
| | 720 | 0.334 | 0.363 | 0.319 | 0.326 | 0.322 | 0.336 | 0.321 | 0.339 | 0.320 | 0.346 | 0.332 | 0.346 |
| | Avg. | 0.207 | 0.263 | 0.359 | 0.275 | 0.235 | 0.270 | 0.238 | 0.275 | 0.241 | 0.283 | 0.242 | 0.279 |
| ECL | 96 | 0.144 | 0.250 | 0.145 | 0.246 | 0.135 | 0.231 | 0.139 | 0.237 | 0.150 | 0.253 | 0.140 | 0.238 |
| | 192 | 0.167 | 0.271 | 0.160 | 0.259 | 0.152 | 0.246 | 0.156 | 0.252 | 0.164 | 0.264 | 0.160 | 0.255 |
| | 336 | 0.194 | 0.294 | 0.182 | 0.278 | 0.173 | 0.267 | 0.175 | 0.270 | 0.181 | 0.282 | 0.180 | 0.276 |
| | 720 | 0.255 | 0.340 | 0.239 | 0.324 | 0.229 | 0.312 | 0.233 | 0.317 | 0.223 | 0.321 | 0.241 | 0.323 |
| | Avg. | 0.190 | 0.288 | 0.182 | 0.277 | 0.172 | 0.264 | 0.176 | 0.269 | 0.180 | 0.280 | 0.180 | 0.273 |
| Traffic | 96 | 0.347 | 0.292 | 0.416 | 0.295 | 0.402 | 0.288 | 0.414 | 0.297 | 0.419 | 0.298 | 0.403 | 0.289 |
| | 192 | 0.398 | 0.305 | 0.424 | 0.306 | 0.416 | 0.294 | 0.426 | 0.301 | 0.434 | 0.305 | 0.415 | 0.296 |
| | 336 | 0.427 | 0.313 | 0.435 | 0.314 | 0.429 | 0.302 | 0.434 | 0.303 | 0.449 | 0.313 | 0.426 | 0.304 |
| | 720 | 0.466 | 0.330 | 0.476 | 0.331 | 0.480 | 0.326 | 0.487 | 0.337 | 0.484 | 0.336 | 0.474 | 0.331 |
| | Avg. | 0.409 | 0.310 | 0.438 | 0.312 | 0.432 | 0.303 | 0.440 | 0.310 | 0.447 | 0.313 | 0.430 | 0.305 |
| 1st Count | | 13 | | 6 | | 21 | | 0 | | 1 | | 2 | |

Table 9: Full short-term time series forecasting. The forecasting horizons are in [6,48] and details in Table 6. Lower values indicate better performance. Red: the best, Blue: second best.

| Methods | | Ours | Time-LLM | GPT4TS | TimesNet | PatchTST | N-BEATS | ETSformer | LightTS | DLinear | FEDformer | Stationary | Autoformer | Informer | Reformer |
|---|---|---|---|---|---|---|---|---|---|---|---|---|---|---|---|
| Yearly | SMAPE | 13.51 | 14.117 | 15.11 | 15.378 | 13.477 | 13.487 | 18.009 | 14.247 | 16.965 | 14.021 | 13.717 | 13.974 | 14.727 | 16.169 |
| | MASE | 3.039 | 3.134 | 3.565 | 3.554 | 3.019 | 3.036 | 4.487 | 3.109 | 4.283 | 3.036 | 3.078 | 3.134 | 3.418 | 3.800 |
| | OWA | 0.796 | 0.827 | 0.911 | 0.918 | 0.792 | 0.795 | 1.115 | 0.827 | 1.058 | 0.811 | 0.807 | 0.822 | 0.881 | 0.973 |
| Quarterly | SMAPE | 10.589 | 11.593 | 10.597 | 10.465 | 10.38 | 10.564 | 13.376 | 11.364 | 12.145 | 11.1 | 10.958 | 11.338 | 11.360 | 13.313 |
| | MASE | 1.262 | 1.424 | 1.253 | 1.227 | 1.233 | 1.252 | 1.906 | 1.328 | 1.520 | 1.35 | 1.325 | 1.365 | 1.401 | 1.775 |
| | OWA | 0.941 | 1.046 | 0.938 | 0.923 | 0.921 | 0.936 | 1.302 | 1.000 | 1.106 | 0.996 | 0.981 | 1.012 | 1.027 | 1.252 |
| Monthly | SMAPE | 13.079 | 14.225 | 13.258 | 13.513 | 12.959 | 13.089 | 14.588 | 14.014 | 13.514 | 14.403 | 13.917 | 13.958 | 14.062 | 20.128 |
| | MASE | 0.984 | 1.101 | 1.003 | 1.039 | 0.97 | 0.996 | 1.368 | 1.053 | 1.037 | 1.147 | 1.097 | 1.103 | 1.141 | 2.614 |
| | OWA | 0.916 | 1.011 | 0.931 | 0.957 | 0.905 | 0.922 | 1.149 | 0.981 | 0.956 | 1.038 | 0.998 | 1.002 | 1.024 | 1.927 |
| Others | SMAPE | 6.435 | 5.125 | 6.124 | 6.913 | 4.952 | 6.599 | 7.267 | 15.880 | 6.709 | 7.148 | 6.302 | 5.485 | 24.460 | 32.491 |
| | MASE | 4.075 | 3.54 | 4.116 | 4.507 | 3.347 | 4.43 | 5.24 | 11.434 | 4.953 | 4.041 | 4.064 | 3.865 | 20.960 | 33.355 |
| | OWA | 1.32 | 1.097 | 1.259 | 1.438 | 1.049 | 1.393 | 1.591 | 3.474 | 1.487 | 1.389 | 1.304 | 1.187 | 5.879 | 8.679 |
| Average | SMAPE | 12.249 | 13.113 | 12.69 | 12.88 | 12.059 | 12.25 | 14.718 | 13.525 | 13.639 | 13.16 | 12.78 | 12.909 | 14.086 | 18.200 |
| | MASE | 1.678 | 1.77 | 1.808 | 1.836 | 1.623 | 1.698 | 2.408 | 2.111 | 2.095 | 1.775 | 1.756 | 1.771 | 2.718 | 4.223 |
| | OWA | 0.89 | 0.946 | 0.94 | 0.955 | 0.869 | 0.896 | 1.172 | 1.051 | 1.051 | 0.949 | 0.930 | 0.939 | 1.230 | 1.775 |

over **14.1%** MSE reductions across all datasets compared to GPT4TS (Zhou et al., 2023). Our improvements are consistently significantly on those typical cross-domain scenarios. For example, our model reduces **5.2%** and **8.9%** MSE compared to best baseline on ETTh1→ ETTh2 and ETTh2→ ETTm2. Significantly, our model exhibits superior size backbone LLM(7B) and is the latest effort in leverage LLMs for zero-shot time series forecasting under "one-to-one" scenarios. We attribute this

Table 10: Full zero-shot learning results on ETT datasets. Lower values indicate better performance.

| Methods | | Ours | | Time-LLM | | GPT4TS | | LLMTime | | PatchTST | | DLinear | |
|---|---|---|---|---|---|---|---|---|---|---|---|---|---|
| Datasets \ Horizon | | MSE | MAE | MSE | MAE | MSE | MAE | MSE | MAE | MSE | MAE | MSE | MAE |
| ETTh1 → ETTh2 | 96 | 0.263 | 0.337 | 0.264 | 0.340 | 0.335 | 0.374 | 0.510 | 0.576 | 0.304 | 0.350 | 0.347 | 0.400 |
| | 192 | 0.315 | 0.374 | 0.332 | 0.376 | 0.412 | 0.417 | 0.523 | 0.586 | 0.386 | 0.400 | 0.447 | 0.460 |
| | 336 | 0.371 | 0.414 | 0.395 | 0.424 | 0.441 | 0.444 | 0.640 | 0.637 | 0.414 | 0.428 | 0.515 | 0.505 |
| | 720 | 0.435 | 0.462 | 0.423 | 0.459 | 0.438 | 0.452 | 2.296 | 1.034 | 0.419 | 0.443 | 0.665 | 0.589 |
| | Avg. | 0.346 | 0.396 | 0.354 | 0.400 | 0.406 | 0.422 | 0.992 | 0.708 | 0.380 | 0.405 | 0.493 | 0.488 |
| ETTh1 → ETTm2 | 96 | 0.191 | 0.296 | 0.224 | 0.311 | 0.236 | 0.315 | 0.646 | 0.563 | 0.215 | 0.304 | 0.255 | 0.357 |
| | 192 | 0.259 | 0.338 | 0.270 | 0.339 | 0.287 | 0.342 | 0.934 | 0.654 | 0.275 | 0.339 | 0.338 | 0.413 |
| | 336 | 0.317 | 0.370 | 0.336 | 0.378 | 0.341 | 0.374 | 1.157 | 0.728 | 0.334 | 0.373 | 0.425 | 0.465 |
| | 720 | 0.409 | 0.424 | 0.410 | 0.422 | 0.435 | 0.422 | 4.730 | 1.531 | 0.431 | 0.424 | 0.640 | 0.573 |
| | Avg. | 0.294 | 0.357 | 0.310 | 0.363 | 0.325 | 0.363 | 1.867 | 0.869 | 0.314 | 0.360 | 0.415 | 0.452 |
| ETTh2 → ETTh1 | 96 | 0.585 | 0.510 | 0.541 | 0.503 | 0.732 | 0.577 | 1.130 | 0.777 | 0.485 | 0.465 | 0.689 | 0.555 |
| | 192 | 0.677 | 0.554 | 0.559 | 0.515 | 0.758 | 0.559 | 1.242 | 0.820 | 0.565 | 0.509 | 0.707 | 0.568 |
| | 336 | 0.700 | 0.562 | 0.620 | 0.551 | 0.759 | 0.578 | 1.328 | 0.864 | 0.581 | 0.515 | 0.710 | 0.577 |
| | 720 | 0.693 | 0.579 | 0.729 | 0.627 | 0.781 | 0.597 | 4.145 | 1.461 | 0.628 | 0.561 | 0.704 | 0.596 |
| | Avg. | 0.663 | 0.551 | 0.612 | 0.549 | 0.757 | 0.578 | 1.961 | 0.981 | 0.565 | 0.513 | 0.703 | 0.574 |
| ETTh2 → ETTm2 | 96 | 0.181 | 0.288 | 0.218 | 0.304 | 0.253 | 0.329 | 0.646 | 0.563 | 0.226 | 0.309 | 0.240 | 0.336 |
| | 192 | 0.235 | 0.324 | 0.265 | 0.335 | 0.293 | 0.346 | 0.934 | 0.654 | 0.289 | 0.345 | 0.295 | 0.369 |
| | 336 | 0.294 | 0.357 | 0.327 | 0.370 | 0.347 | 0.376 | 1.157 | 0.728 | 0.348 | 0.379 | 0.345 | 0.397 |
| | 720 | 0.395 | 0.411 | 0.401 | 0.416 | 0.446 | 0.429 | 4.730 | 1.531 | 0.439 | 0.427 | 0.432 | 0.442 |
| | Avg. | 0.276 | 0.345 | 0.303 | 0.356 | 0.335 | 0.370 | 1.867 | 0.869 | 0.325 | 0.365 | 0.328 | 0.386 |
| ETTm1 → ETTh2 | 96 | 0.415 | 0.437 | 0.331 | 0.383 | 0.353 | 0.392 | 0.510 | 0.576 | 0.354 | 0.385 | 0.365 | 0.415 |
| | 192 | 0.486 | 0.477 | 0.353 | 0.399 | 0.443 | 0.437 | 0.523 | 0.586 | 0.447 | 0.434 | 0.454 | 0.462 |
| | 336 | 0.397 | 0.433 | 0.400 | 0.428 | 0.469 | 0.461 | 0.640 | 0.637 | 0.481 | 0.463 | 0.496 | 0.494 |
| | 720 | 0.451 | 0.475 | 0.417 | 0.448 | 0.466 | 0.468 | 2.296 | 1.034 | 0.474 | 0.471 | 0.541 | 0.529 |
| | Avg. | 0.437 | 0.455 | 0.375 | 0.415 | 0.433 | 0.439 | 0.992 | 0.708 | 0.439 | 0.438 | 0.464 | 0.475 |
| ETTm1 → ETTm2 | 96 | 0.081 | 0.185 | 0.194 | 0.270 | 0.217 | 0.294 | 0.646 | 0.563 | 0.195 | 0.271 | 0.221 | 0.314 |
| | 192 | 0.162 | 0.250 | 0.243 | 0.304 | 0.277 | 0.327 | 0.934 | 0.654 | 0.258 | 0.311 | 0.286 | 0.359 |
| | 336 | 0.250 | 0.311 | 0.295 | 0.341 | 0.331 | 0.360 | 1.157 | 0.728 | 0.317 | 0.348 | 0.357 | 0.406 |
| | 720 | 0.376 | 0.392 | 0.367 | 0.385 | 0.429 | 0.413 | 4.730 | 1.531 | 0.416 | 0.404 | 0.476 | 0.476 |
| | Avg. | 0.217 | 0.284 | 0.275 | 0.325 | 0.313 | 0.348 | 1.867 | 0.869 | 0.296 | 0.334 | 0.335 | 0.389 |
| ETTm2 → ETTh2 | 96 | 0.485 | 0.473 | 0.322 | 0.369 | 0.360 | 0.401 | 0.510 | 0.576 | 0.327 | 0.367 | 0.333 | 0.391 |
| | 192 | 0.496 | 0.481 | 0.359 | 0.396 | 0.434 | 0.437 | 0.523 | 0.586 | 0.411 | 0.418 | 0.441 | 0.456 |
| | 336 | 0.542 | 0.509 | 0.439 | 0.452 | 0.460 | 0.459 | 0.640 | 0.637 | 0.439 | 0.447 | 0.505 | 0.503 |
| | 720 | 0.437 | 0.463 | 0.448 | 0.468 | 0.485 | 0.477 | 2.296 | 1.034 | 0.459 | 0.470 | 0.543 | 0.534 |
| | Avg. | 0.490 | 0.481 | 0.392 | 0.421 | 0.435 | 0.443 | 0.992 | 0.708 | 0.409 | 0.425 | 0.455 | 0.471 |
| ETTm2 → ETTm1 | 96 | 0.345 | 0.377 | 0.446 | 0.415 | 0.747 | 0.558 | 1.179 | 0.781 | 0.491 | 0.437 | 0.570 | 0.490 |
| | 192 | 0.518 | 0.462 | 0.496 | 0.452 | 0.781 | 0.560 | 1.327 | 0.846 | 0.530 | 0.470 | 0.590 | 0.506 |
| | 336 | 0.735 | 0.541 | 0.507 | 0.463 | 0.778 | 0.578 | 1.478 | 0.902 | 0.565 | 0.497 | 0.706 | 0.567 |
| | 720 | 0.653 | 0.533 | 0.556 | 0.482 | 0.769 | 0.573 | 3.749 | 1.408 | 0.686 | 0.565 | 0.731 | 0.584 |
| | Avg. | 0.562 | 0.478 | 0.501 | 0.453 | 0.769 | 0.567 | 1.933 | 0.984 | 0.568 | 0.492 | 0.649 | 0.537 |
| 1st Count | | 21 | | 16 | | 0 | | 0 | | 8 | | 0 | |

success to our multi-level text alignment being better at activating the LLM's knowledge transfer and reasoning capabilities in a resource-efficient manner when performing time series tasks.

