# OpenReview forum: "Enhancing LLM's interpretability for time series via multi-level aligned embeddings"
_ICLR.cc/2025/Conference — ICLR 2025 Conference Withdrawn Submission_

### Official Review · Reviewer_AEVd · 2024-10-17

**Soundness:** 3
**Presentation:** 2
**Contribution:** 3
**Rating:** 5
**Confidence:** 4

**Summary:**

The paper proposes a multi-level text alignment framework to enhance the interpretability and performance of LLMs in time series forecasting. It decomposes time series data into trend, seasonal, and residual components, aligning each with specific text representations to improve interpretability while maintaining predictive accuracy. Additionally, component-specific prompts guide the generation of continuous vector representations to capture temporal knowledge. Experimental results show that this method outperforms state-of-the-art models across multiple datasets, demonstrating its effectiveness in both accuracy and interpretability.

**Strengths:**

1. This paper concerns an interesting and promising problem in our field, ensuring the semantic consistency between text prototypes of LLM and the features (such as trends) of time series.

2. The method proposed in this paper balances prediction accuracy and interpretability by decomposing the original time series, aligning components with different word pools, and predicting them individually.

3. The extensive experiments look hopeful.

**Weaknesses:**

1. **Principal ambiguity.** This paper aims to enhance the interpretability of LLMs for time series, specifically ensuring semantic consistency between textual prototypes in the LLM and features (e.g., trends) in the time series. While the authors present promising experimental results in Figure 3, I still find it unclear how their algorithm achieves this. Compared to S2IP-LLM, their designs are incremental:

    - Each component after the decomposition interacts with a different word pool;
    - The interaction transforms from concat to cross-attention;
    - Three  component-specific prompts are packed with different components;
    - Each component is predicted separately.

    The last two steps aim to enhance prediction accuracy, while the first two may relate to interpretability. However, it remains challenging to understand how consistency is ensured merely through unconstrained cross-attention between the trend component and the trend-specific word pool. Why do patches with upward trends naturally align with words signifying upward trends, not words indicating downward trends? Can the author provide a more detailed explanation or even an anonymous code repository? Additionally, a minor question: Are the three component-specific word pools the same across different datasets?

2. **Writing errors.** The main text lacks references to Figure 1 and Table 9.

If the authors address my questions and concerns, I will reconsider raising my score.

**Questions:**

The authors can refer to the weakness listed above.

---

### Official Review · Reviewer_qXBM · 2024-10-31

**Soundness:** 3
**Presentation:** 3
**Contribution:** 2
**Rating:** 5
**Confidence:** 3

**Summary:**

The paper proposes a multi-level text alignment framework for adapting large language models (LLMs) to time series forecasting. Recognizing the challenges posed by the continuous nature of time series data, the authors introduce a method to transform time series components (trend, seasonal, and residual) into component-specific text representations, aligning these with pre-trained language embeddings. This multi-level alignment aims to enhance the interpretability and predictive accuracy of LLMs for time series forecasting by leveraging the reasoning abilities of LLMs in a new context. The paper's key contributions include: (1) a decomposition approach that separates time series into interpretable components aligned with text representations; (2) the development of component-specific prompts and embeddings to guide the model in generating meaningful representations of temporal patterns; and (3) empirical validation across various datasets, where the proposed method demonstrates superior performance over traditional and LLM-based time series forecasting models, particularly in long-term and few-shot forecasting tasks.

**Strengths:**

- **Clear Methodology**: The methodology is well-structured and easy to follow, with each step clearly explained. The process, from decomposing time series into trend, seasonal, and residual components to aligning them with interpretable embeddings, is accessible and logically presented, supported by clear diagrams.
- **Extensive Experimentation**: The paper includes a comprehensive set of experiments on various benchmark datasets, covering a range of forecasting horizons. This thorough experimentation supports the model's robustness and adaptability across diverse time series patterns, adding credibility to its claims.
-  **Strong Experimental Results**: The model demonstrates notable improvements in predictive accuracy across multiple forecasting tasks, including long-term, short-term, few-shot, and zero-shot scenarios. The results indicate the approach’s effectiveness in enhancing accuracy and interpretability over conventional and LLM-based forecasting models.

**Weaknesses:**

- **Missing Error Bounds**: The paper lacks error bounds. Running experiments multiple times and reporting ranges in Table 1 would provide a clearer assessment of performance stability.
- **Single-Model Limitation**: The evaluation is restricted to GPT-2. Testing the method on other LLMs would better demonstrate its effectiveness.
- **Interpretability of Seasonal and Residual Components**: Interpretability claims are stronger for trends than for seasonal and residual components. Adding case studies or visualizations for these components would better support the interpretability claims.
- **Computational Overhead**: The paper does not address the computational cost of the multi-level alignment framework. An analysis of training and inference efficiency, especially on large datasets, would clarify the method’s scalability and practicality.

**Questions:**

- **Clarification on Terminology – LLM Interpretability**: The term “LLM interpretability” seems to obscure the specific aim of aligning time series with text embeddings. Wouldn’t “TS-text alignment” be a more accurate term, focusing directly on the technique rather than implying broader interpretability of LLMs? Could the authors clarify this choice and how it impacts the framing of their contributions?
- **Choice of GPT-2 and Generalization Across LLM Architectures**: Given that only GPT-2 was used, there is limited evidence that this method generalizes across different LLMs. Why was GPT-2 chosen as the primary model, and what justifications are there for expecting similar performance on larger or more recent LLMs? Could the authors address potential variations in results due to differences in training objectives and capacities of other LLM architectures?
- **Use of Only GPT-2’s First 6 Layers**: The paper utilizes only the first 6 layers of GPT-2, which raises questions about whether the approach truly “adapts” an LLM for time series forecasting. Six layers alone do not embody the text generation capabilities typically associated with LLMs, suggesting a limited adaptation. Could the authors justify how this partial model use aligns with the goals of leveraging LLMs for interpretability and predictive accuracy?

---

### Official Review · Reviewer_vRLz · 2024-11-01

**Soundness:** 2
**Presentation:** 2
**Contribution:** 2
**Rating:** 3
**Confidence:** 5

**Summary:**

This paper presents a novel approach to enhance the interpretability of using LLM for time series forecasting. It introduces a multi-level text alignment framework that decomposes time series into trend, seasonal, and residual components. Then the time series components are encoded into text representations aligned with pre-trained word tokens. This method aims to improve the accuracy and interpretability of time series forecasting.

**Strengths:**

1. The mechanism to align each decomposed time series representation with word representation is unique.
2. By aligning time series components with language-based representations, the model enhances forecast interpretability, making it easier for users to understand the reasoning behind the model's predictions.
3. The experiment has a rich setting and comprehensive comparison with state-of-the-art baselines. The performance validates the effectiveness of the approach.

**Weaknesses:**

1. Concerning using the LOESS method for decomposing the entire time series X(i) ∈ R^1×L. There is a potential risk of data leakage that could inadvertently lead to favorable performance. It would be beneficial to elucidate your methodology when implementing this technique.
2. Given the LLM is frozen, it is better to try different LLM backbones (LLaMA, BERT, etc.) and report how the performance and interpretability of their method varies across different model architectures.
3. The claim of the interpretability of time series is too broad. Firstly, Showing highlights maps of trend components with selected words is insufficient. Second, there is limited exploration of residuals and seasonality.

**Questions:**

See weakness.

---

### Note · Authors · 2024-11-25

I have read and agree with the venue's withdrawal policy on behalf of myself and my co-authors.